# Decentralized Byzantine-Resilient Multi-Agent Reinforcement Learning with Reward Machines in Temporally Extended Tasks

## Abstract

In resilient cooperative multi-agent reinforcement learning (c-MARL), a fraction of agents exhibit Byzantine behavior, sending fabricated or adversarially crafted information to hinder the learning process. Unlike existing approaches that often rely on a central controller or impose stringent behavior requirements on agents, we propose a fully decentralized method using reward machines (RMs) that can learn an optimal policy for temporally extended tasks. We introduce a belief-based Byzantine detection mechanism for discrete-time multi-agent reinforcement learning (MARL), where defender (non-Byzantine) agents iteratively update probabilistic suspicions of peers using observed actions and rewards. RMs allow us to encode the temporal dependencies in the reward structure of the task and guide the learning process. Our methods introduce tabular Q-learning and actor-critic algorithms with reward machines to learn a robust consensus mechanism to isolate the influence of Byzantine agents, in order to ensure effective learning by defender agents. We establish theoretical guarantees, demonstrating that our algorithms converge to an optimal policy. We further evaluate our method against baselines in two case studies to show its effectiveness and performance.

## 1 Introduction

In multi-agent settings, where multiple learners interact within a shared environment, reinforcement learning (RL) enables collaborative exploration and policy optimization. However, real-world deployments, such as autonomous vehicle networks or distributed sensor arrays often involve distributed systems where agents communicate locally. One main challenge in these settings arises from Byzantine agents, which may act arbitrarily or maliciously due to adversarial attacks or malfunction Lamport & Fischer (1982); Yin et al. (2018), transmitting corrupted data to undermine the collective learning objective.

RL frameworks assume benign agents or centralized coordination, rendering them vulnerable to such adversarial disruptions. Recent advances, such as those in distributed RL Alsadat et al. (2025), have begun addressing robustness, yet many rely on a central server. This work tackles Byzantine-robust multi-agent RL in a fully decentralized setting, where agents communicate over a time-varying network without a central coordinator. We introduce two novel algorithms: BQL-RM (Belief-based Q-learning with Reward Machines) and BAC-RM (Belief-based Actor-Critic with Reward Machines) for resilient multi-agent reinforcement learning (MARL).

In our methods, we use reward machines (RMs) Icarte et al. (2018), which are a type of Mealy machine that encode the task structure, and provide a structured approach to specify tasks with Markovian or non-Markovian reward functions for reinforcement learning agents. RMs provide a formal framework for representing complex Alsadat & Xu (2024), temporally extended tasks in a way that can guide reinforcement learning similar to temporal logic Aria & Xu (2025), but with the added advantage of learnable temporal dependencies and allows for transfer learning since it learns the task structure Icarte et al. (2018). By encoding the reward structure as a finite state machine, RMs make the temporal dependencies in the task explicit and learnable. This is particularly important for tasks that depend on temporal structure of the events or have long-term dependencies that are difficult to capture with standard Markovian reward functions.

**Related Work.** In resilient RL, there are several main approaches to handle adversarial agents. One is to use robust aggregation Huang et al. (2024); Blanchard et al. (2017), another is to use a belief mechanism for Byzantine detection Li et al. (2023) or considering adversarial agents as uncertainty in the environment He et al. (2023). Our approach combines decentralized belief updates with RMs to handle temporally extended tasks without a central coordinator.

**Byzantine-Robust Distributed RL:** Byzantine resilience in distributed systems traces to Lamport's seminal work on fault-tolerant consensus Lamport & Fischer (1982). Recent RL adaptations, such as Byzan-UCBVI Zhang et al. (2021) requires episodic synchronization, incurring high communication costs. On the other hand, clique-overweight (COW) Chen et al. (2023), proposes robust mean estimators for aggregating gradients from untrusted batches. While COW handles arbitrary batch sizes, it focuses on supervised settings and assumes a central server; a single point of failure in decentralized MARL. In contrast, our belief mechanism operates fully decentralized.

**Adversarial MARL:** Adversarial methods like RADAR Phan et al. (2021) co-train the protagonist and antagonist agents, but require exhaustive adversary sampling, becoming intractable for large systems. M3DDPG Li et al. (2019) uses minimax optimization for worst-case perturbations but assumes fixed budgets, failing against adaptive adversaries. BARDec-POMDP Li et al. (2023) frames adversaries as Bayesian types but lacks convergence guarantees. COMA Foerster et al. (2018) and PPO-QMIX Rashid et al. (2020) excel in cooperation but require fixed agent numbers. Our work eliminates ratio assumptions and provides rigorous discrete-time analysis with convergence guarantees.

**Belief-Based Coordination:** Belief systems in ad hoc teamwork Stone et al. (2010); Albrecht & Ramamoorthy (2015) enable agents to adapt to unknown teammate types. However, these methods presume cooperative agents with shared objectives, unlike our adversarial setting. Recent extensions Rahman et al. (2021) handle open teams (agents can enter and leave the team), but do not address Byzantine failures. Closest to our work is Tessler et al. (2018), which uses reward shaping for robustness, but their mechanism lacks theoretical grounding.

We present a belief-based framework that uses reward machines to encode task structure and guide learning in cooperative MARL. We further propose a robust Byzantine-detection mechanism over an augmented state space with provable guarantees. Using these components, we develop two algorithms, a Q-learning approach and an actor-critic method that integrate RMs with our detection scheme. We provide theoretical proofs establishing convergence to an optimal policy.

## 2 PROBLEM FORMULATION

We model the environment as a multi-agent labeled Markov decision process (MDP).

**Labeled Markov Decision Processes.** We define a multi-agent labeled MDP as a tuple $\mathcal{M} = (\mathcal{S}, \mathcal{N}, \boldsymbol{s}_I, A, p, \gamma, \mathcal{P}, L)$ which consists of a finite set of states denoted by $\mathcal{S}$, a finite set of agents denoted by $\mathcal{N}$ such that $\mathcal{N} = \{1, \ldots, \mathbb{N}\}$ where $\mathbb{N} \in \mathbb{N}$ is the total number of agents, initial states denoted by $\boldsymbol{s}_I = \{s_I^0, \ldots, s_I^{\mathbb{N}}\}$ where for each agent $i \in \mathcal{N}$ we have $s_I^i \in \mathcal{S}$, a finite set of joint actions denoted by $A = \prod_{i \in \mathcal{N}} A^i$, a transition probability function denoted by $P : \mathcal{S} \times A^i \times \mathcal{S} \to [0, 1]$, a discount factor denoted by $\gamma \in [0, 1)$, a finite set of atomic propositions denoted by $\mathcal{P}$, and a labeling function denoted by $L : \mathcal{S} \times A^i \times \mathcal{S} \to 2^{\mathcal{P}}$ that maps transitions to sets of propositions.

Reward machines (RMs) are a type of Mealy machine that encode Markovian and non-Markovian reward functions. They process a sequence of labels to produce rewards.

**Definition 1 (Reward machine)** *A reward machine (RM) is defined as* $\mathcal{A} = \langle \mathbb{U}, u_I, 2^{\mathcal{P}}, M, \delta, \sigma \rangle$ *where* $\mathbb{U}$ *is a finite set of states,* $u_I \in \mathbb{U}$ *is the initial state,* $2^{\mathcal{P}}$ *is the input alphabet,* $M \subseteq \mathbb{R}$ *is the output alphabet (which represents its reward),* $\delta : \mathbb{U} \times 2^{\mathcal{P}} \mapsto \mathbb{U}$ *is the transition function, and* $\sigma : \mathbb{U} \times 2^{\mathcal{P}} \mapsto M$ *is the output function mapping to reward.*

An RM $\mathcal{A}^i$ for agent $i$ processes a label sequence $l_0^i l_1^i \ldots l_k^i$ to generate a sequence of RM states $u_0^i u_1^i \ldots u_k^i$ and a corresponding reward sequence $r_0^i r_1^i \ldots r_k^i$. The initial RM state is $u_0^i = u_I^i$, and subsequent states are determined by $u_{k+1}^i = \delta^i(u_k^i, l_{k+1}^i)$. Rewards are computed as $r_k^i = \sigma^i(u_k^i, l_{k+1}^i)$. The reward function of the labeled MDP for agent $i$ is replaced with the reward ma-

chine $\mathcal{A}^i$ that maps a trajectory $s_0^i a_0^i s_1^i a_1^i \ldots s_k^i a_k^i s_{k+1}^i$ with labels $l_0^i l_1^i \ldots l_k^i$ to the reward sequence $r_0^i r_1^i \ldots r_k^i$, as defined by $\mathcal{A}^i(l_0^i l_1^i \ldots l_k^i)$.

A policy $\pi^i(s^i, u^i, a^i)$ for agent $i$ defines the probability of taking action $a^i \in A^i$ in MDP state $s^i \in \mathcal{S}$ given RM state $u^i \in \mathbb{U}$. Under policy $\pi^i$, agent $i$ in MDP state $s^i$ and RM state $u^i$ reaches MDP state $s'$ and RM state $u'$ with probability $P(s^i, u^i, a^i, s^{i,'}, u^{i,'})$ after taking action $a^i$. We denote a trajectory by $s_0^i u_0^i a_0^i s_1^i u_1^i a_1^i \ldots s_k^i u_k^i a_k^i s_{k+1}^i u_{k+1}^i$ where $k \in \mathbb{N}$ represents a sequence of states and actions visited by agent $i$. The corresponding label sequence to a trajectory is $\lambda^i = l_0 l_1 \ldots l_k$ where $l_k = L(s_k^i, a_k^i, s_{k+1}^i)$. A trace is a pair $(\lambda^i, \rho^i)$ consisting of a label sequence $\lambda^i$ and corresponding reward sequence $\rho^i = r_0^i r_1^i \ldots r_k^i$.

## 3 METHODOLOGY

In cooperative multi-agent systems, it may be the case that some of the agents are improvised or malicious, i.e., they are not following the specified policy either due to a malfunction or adversarial attacks; therefore, experience Byzantine failures Yin et al. (2018); Xue et al. (2021). We characterize the Byzantine agents by their type $\theta^i \in \{0, 1\}$, 0 is for defender agents and 1 is for Byzantine agents. We denote the finite set of types by $\Theta = \times_{i \in \mathcal{N}} \theta^i$. In our setting, we assume that the Byzantine agents can send fabricated or adversarially crafted or adversarially crafted information to hinder the learning process of defender agents. At the beginning of each episode, we randomly select a fraction of agents to be Byzantine, i.e., $\theta^i = 1$ for $i \in \mathcal{N}_B$ where $\mathcal{N}_B \subseteq \mathcal{N}$ is the set of Byzantine agents. We apply Assumption 1 to ensure that the number of Byzantine agents is less than the number of defenders, so that the defender can learn a robust policy against Byzantine agents.

**Assumption 1 (Byzantine agents number)** *We apply the constraint to the number of Byzantine agents, i.e., $|\mathcal{N}| - |\mathcal{N}_B| \geq M|\mathcal{N}_B| + 1$ where $M \in \mathbb{Z}^+ \setminus \{0\}$ ensures that the number of defender agents is at least $M$ times the number of Byzantine agents.*

For the Byzantine agents, we assume that they can send fabricated or adversarially crafted or adversarially crafted information to hinder the learning process of defender agents, namely, we characterize this attack in our setting by sampling an action from the Byzantine agent's policy $\hat{\pi}^i(\cdot \mid s^i, u^i, b^i)$, i.e., $\hat{a}_k^i \sim \hat{\pi}^i(\cdot \mid s^i, u^i, b^i)$ and replacing the defender agent's action $a_k$ with $\hat{a}_k^i$. Therefore, the Byzantine agents at time $k$ can send messages to the defender agents, where the defender agents can use these messages to update their policy. Agents execute the actions simultaneously and transition to the next state with probability $p(s_{k+1}^i, u_{k+1}^i, b_{k+1}^i \mid s_k^i, u_k^i, b_k^i, \hat{a}_k^i)$. By executing the Byzantine agent's action, the defender agents can be misled to learn a suboptimal policy. We assume that the Byzantine agents can send fabricated information to the defender agents. This assumption is common in Byzantine-robust RL literature Chen et al. (2023); Zhang et al. (2021). We can write the value function for each type as follows:

$$V_{\theta^i}(s^i, u^i, b^i) = \mathbb{E}_{\hat{a}_k^i \sim \hat{\pi}^{\theta^i}(\cdot \mid s_k^i, u_k^i, b_k^i)} \left[ \sum_{k=0}^{\infty} \gamma^k r_k^i \right] \tag{1}$$

Across the current literature, some methods assume that the Byzantine agents can only send adversarial information to the defender agents, such as M3DDPG Li et al. (2019) and ROMAX Sun et al. (2022). Our method also uses state-adversarial MDP Zhang et al. (2020) since it considers the adversary during the decision-making process.

### 3.1 ADVERSARIAL ATTACK MODEL

There have been studies for single-agent and multi-agent reinforcement learning where adversarial attacks are in the form of action perturbations Tessler et al. (2019); Li et al. (2019). Moreover, authors in Li et al. (2019) extend the action perturbation model to multi-agent reinforcement learning by offering a more dynamic and a less conservative alternative to existing methods by treating agent types as uncertain and using belief updates while remaining robust against adversarial attacks. Action uncertainties, often modeled as adversarial attacks like adversarial policies Gleave et al. (2019);

Wu et al. (2021); Guo et al. (2021) or non-oblivious adversaries Dinh et al. (2023), represent a practical and disruptive form of attack that is difficult to mitigate. Building upon these works, we propose a realistic threat model with specific assumptions regarding attackers and defenders.

**Assumption 2 (Adversarial Model)** *During an episode, a fraction of agents are Byzantine, i.e., $\theta^i = 1$ for $i \in \mathcal{N}_B$, and they can send fabricated information to the defender agents. The defender agents are unaware of the types of other agents and must learn a robust policy against the worst-case adversary. The type $\theta^i$ is determined by nature and cannot be changed during an episode Li et al. (2023).*

In our work, there can be more than one Byzantine agent as long as Assumption 1 holds. Despite similar methods such as Li et al. (2019), we assume that in each episode, more than one agent could be Byzantine. Additionally, the type space may be more complex, having a non-binary type space Xie et al. (2022), perturbing actions irregularly Lin et al. (2017). In a resilient cooperative multi-agent reinforcement learning setting with fixed policies, there exists a worst-case adversary that can cause the most harm to the defender agents Li et al. (2023). We also assume limitations in defender agents' capabilities and information access during the training.

**Assumption 3 (defenders' limitations)** *We assume that the defender agents can communicate with each other only through local communication and can only observe their own actions and rewards and their neighbors' actions and rewards. The defender agents do not have access to the type $\theta^i$ of other agents. The defender agents can only observe the labels associated with their own actions. We also assume that the policy of defender agents is fixed against the Byzantine agents' policies, $\hat{\pi}^{i,\star}$.*

### 3.2 Belief-based Byzantine Detection

In this section, we introduce our belief-based mechanism for detecting Byzantine agents. Each agent $i \in \mathcal{N}$ maintains a belief state $b_j^i \in \mathsf{B}$ where $\mathsf{B}$ is a finite set of beliefs for each of its neighbors $j \in \mathcal{N}$. This value represents agent $i$'s belief that neighbor $j$ is Byzantine. At the start of each episode, the belief state is initialized to a prior value $p \in (0,1)$. It is then updated iteratively based on the observed actions and rewards of the neighbors, following the update rules described below.

$$\zeta_j^i(k+1) = \begin{cases} \zeta_j^i(k) + \gamma^+ \alpha(1 - \zeta_j^i(k)) & \text{if } a_k^j \neq a_k^{\star,j}, \\ \zeta_j^i(k) - \gamma^- \alpha \zeta_j^i(k) & \text{if } a_k^j = a_k^{\star,j}, \end{cases} \quad (2)$$

If the neighbor agent $j$ takes a non-optimal action (by comparing it against its own optimal action in the same state), that is, $a_k^j \neq a_k^{\star,j}$, the belief state is increased by a factor of $\gamma^+ \alpha(1 - b_j^i(k))$, where $\gamma^+ > 0$ is the update rate for non-optimal actions and $\alpha \in (0,1)$ is the learning rate. Conversely, if the neighbor agent $j$ takes an optimal action, i.e., $a_k^j = a_k^{\star,j}$, the belief state is decreased by a factor of $\gamma^- \alpha b_j^i(k)$, where $\gamma^- > 0$ is the update rate for optimal actions. After each update, the belief state is discretized into three categories: if $b_j^i(k+1) \leq \beta_l$, the agent $i$ considers the neighbor agent $j$ as a defender, i.e., $\mathcal{B}_j^i = 0$; if $\beta_l < b_j^i(k+1) \leq \beta_u$, the agent $i$ considers the neighbor agent $j$ as suspicious , i.e., $\mathcal{B}_j^i = 1$; and if $b_j^i(k+1) > \beta_u$, the agent $i$ considers the neighbor agent $j$ as Byzantine, i.e., $\mathcal{B}_j^i = 2$ where $\beta_l, \beta_u \in [0,1]$ and $\beta_l < \beta_u$ are the lower and upper thresholds for the belief state, respectively. Theorem 1 shows that the belief update mechanism converges.

**Theorem 1 (Belief Update Convergence)** *The belief update mechanism in equation 2 converges to the ground truth belief state, where each agent $i \in \mathcal{N}$ maintains a belief state $b_j^i \in \mathsf{B}$ for each of its neighbors $j \in \mathcal{N}$.*

Proof of Theorem 1 is provided in the Appendix.

Algorithm 1 shows how each agent updates its belief state to detect Byzantine agents. Each agent $i$ maintains a belief $b^i$ about which other agents may be Byzantine, based on observed actions and rewards. The algorithm allows agents to identify and isolate those sending fabricated or faulty information. Each agent maintains a probabilistic belief about whether the other agents are Byzantine, updating these beliefs iteratively based on observed actions and rewards. The algorithm begins by initializing the belief states for each agent $i \in \mathcal{N}$ at time $k = 0$, where for every agent

---

**Algorithm 1** Belief update for detecting Byzantine agents

---

**Input**: Agent set $\mathcal{N}$, initial belief prior $p \in (0, 1)$
**Parameter**: learning rate $\alpha \in (0, 1)$, suspicion threshold $\beta \in (0, 1)$, update rates $\gamma^+, \gamma^-, \gamma_r > 0$
**Output**: $\{\zeta_j^i\}, \{i, j \in \mathcal{N}\}$
1: **Function BeliefUpdate()**
2: **for** $i \in \mathcal{N} \wedge k = 0$ **do**
3:    **for** $j \in \mathcal{N}, j \neq i$ **do**
4:       $\zeta_j^i(0) \leftarrow p$
5:    **end for**
6:    $\zeta_i^i(0) \leftarrow 0$
7: **end for**
8: **for** $i \in \mathcal{N}$ **do**
9:    **for** $j \in \mathcal{N} \wedge j \neq i$ **do**
10:      Observe $a_k^j$ and $r_k^j$
11:      Compute optimal action $a_k^{\star, j}$
12:      *// Action-based belief update (Equation (2))*
13:      **if** $a_k^j \neq a_k^{\star, j}$ **then**
14:         $\zeta_j^i(k+1) \leftarrow \zeta_j^i(k) + \gamma^+ \alpha(1 - \zeta_j^i(k))$
15:      **else**
16:         $\zeta_j^i(k+1) \leftarrow \zeta_j^i(k) - \gamma^- \alpha \zeta_j^i(k)$
17:      **end if**
18:      *// Discretize belief state*
19:      **if** $\zeta_j^i(k+1) \leq \beta_l$ **then**
20:         $\mathcal{B}_j^i \leftarrow 0$ *// Defender*
21:      **else if** $\beta_l < \zeta_j^i(k+1) \leq \beta_u$ **then**
22:         $\mathcal{B}_j^i \leftarrow 1$ *// Suspicious*
23:      **else**
24:         $\mathcal{B}_j^i \leftarrow 2$ *// Byzantine*
25:      **end if**
26:    **end for**
27: **end for**
28: **return** $\{\zeta_j^i\}, \{i, j \in \mathcal{N}\}$

---

$j \neq i$, agent $i$ assigns an initial belief probability $\zeta_j^i(0) = p$, with $p \in (0, 1)$ as a prior (Lines 2 to 4), and sets its belief about itself to zero, i.e., $\zeta_i^i(0) = 0$ (Line 6). We then iterate over all agent pairs $i$ and $j \neq i$ at each time step (Lines 8 to 9) to update belief states for each $i$ and $j$ at time step $k$. Agent $i$ observes $j$'s action $a_k^j$ (sampling action from Byzantine agents Li et al. (2023)) and reward $r_k^j$ (Line 10), and then, using its own value function, each agent $i$ determines whether the action taken by another agent $j$ was optimal (Line 11). If the observed action differs from the optimal (Line 13), the belief is increased (i.e., the suspicion that the agent $j$ is Byzantine increases) via $\zeta_j^i(k+1) \leftarrow \zeta_j^i(k) + \gamma^+ \alpha(1 - \zeta_j^i(k))$ (Line 14); otherwise, it decreases via $\zeta_j^i(k+1) \leftarrow \zeta_j^i(k) - \gamma^- \alpha \zeta_j^i(k)$ (Line 16). The belief is then discretized into defender (0), suspicious (1), or Byzantine (2) using thresholds for lower and upper bounds, i.e., $\beta_l$ and $\beta_u$ (Lines 19 to 24), assigning $\mathcal{B}_j^i$ accordingly. Finally, the algorithm returns the set of beliefs probabilities $\{\zeta_j^i\}$ for all agent pairs (Line 28), enabling robust decision-making in the presence of Byzantine agents.

### 3.3 Q-LEARNING WITH REWARD MACHINES AND BELIEF STATES

In this section, we present our Q-learning algorithm that incorporates reward machines and belief states to learn robust policies in the presence of Byzantine agents. We extend the standard Q-learning algorithm to handle reward machines and belief states. The Q-learning algorithm learns a Q-function $Q^i(s^i, u^i, b^i, a^i)$ for each agent $i \in \mathcal{N}$, which represents the expected cumulative reward for taking action $a^i$ in state $s^i$ with RM state $u^i$ and belief state $b^i$. The Q-function is updated using the Bellman equation shown in Definition 3.

We also incorporate the belief update mechanism from Algorithm 1 to update the belief states of the agents based on the observed actions and rewards of their neighbors.

**Definition 2 (Q-function)** *The Q-function for agent $i$ is defined as $Q^i(s^i, u^i, b^i, a^i)$, which represents the expected cumulative reward for taking action $a^i$ in state $s^i$ with RM state $u^i$ and belief state $b^i$.*

**Definition 3 (Q-learning update)** *The Q-learning update for agent $i$ is defined as:*

$$
\begin{aligned}
Q^i(s^i, u^i, b^i, a^i) \leftarrow (1 - \alpha_Q) \cdot Q^i(s^i, u^i, b^i, a^i) + \\
\alpha_Q \cdot (r_k^i + \gamma \cdot \max_{a^{i,\prime} \in A^i} Q^i(s^{i,\prime}, u^{i,\prime}, b^{i,\prime}, a^{i,\prime}))
\end{aligned}
\tag{3}
$$

*where $\alpha_Q \in (0, 1)$ is the learning rate, $\gamma \in [0, 1)$ is the discount factor, $s^{i,\prime}$ is the next MDP state, $u^{i,\prime}$ is the next RM state, and $b^{i,\prime}$ is the next belief state.*

We use Algorithm 2 to train agents using Q-learning with reward machines and belief states. The proposed algorithm, BQL-RM (belief-based Q-learning with reward machines), is designed to learn decentralized optimal policies in cooperative multi-agent systems with Byzantine agents. First, we initialize the algorithm by initializing the Q-function $Q^i(s^i, u^i, b^i, a^i)$ for each agent $i \in \mathcal{N}$ across all states $s^i$, RM states $u^i$, belief states $b^i$, and actions $a^i$ (Line 2). Then we set the initial RM state to $u_0^i = u_I$ (Line 3), and initializes belief states $b_0^i$ (Line 3). Afterwards, we iterate over $\mathcal{Q}$ episodes (Line 5) where we reset the environment state to $s_0^i$ (Line 6), reinitialize RM states (Line 7), and resets belief states using Algorithm 1. Within each episode, a time step loop from $k = 0$ to $T$ (Line 9) initializes action and belief lists (Line 10), selects actions via an epsilon greedy policy (Line 13), and updates lists (Line 14). Then each agent executes its action and transitions to a new MDP state $s_{k+1}^i$ (Line 18) and receives a label $l_{k+1}^i$ from the environment (Line 19). By receiving the label, the agent then updates its RM state (Line 21) and receives a reward $r_k^i$ (Line 21). We then update the belief states every $m \in \mathbb{Z}_+$ steps (Lines 23 to 25) in order to find and isolate the Byzantine agent. Afterwards, the Q-values are updated using equation 3 to learn the optimal action-value functions (Line 31). We then transition to the next time step, following the state updates (Line 32). Finally, the BQL-RM returns the Q-functions $\{Q^i\}_{i \in \mathcal{N}}$ (Line 36), encoding optimal action-value functions for each agent's task, in the adversarial context, ensuring robust learning against Byzantine agents.

## 4 ALGORITHMS

We analyze the convergence properties of BQL-RM and BAC-RM, establishing theoretical guarantees for learning in Byzantine environments. BQL-RM converges to optimal policies under tabular assumptions, while BAC-RM converges to stationary points with function approximation. Detailed proofs are in the appendix.

### 4.1 LEARNING GUARANTEES FOR BQL-RM

BQL-RM convergence analysis extends classical Q-learning results to augmented state spaces with Byzantine agents and belief updates. First, we require finite state spaces $\mathcal{S}$, U, and B to ensure finite Q-table dimensions, which is practical since reward machines have finite states and belief spaces can be discretized.

Second, we require that all state-action pairs $(s^i, u^i, b^i, a^i)$ are visited infinitely often with probability 1. This exploration condition can be satisfied through appropriate $\epsilon$-greedy policies with $\epsilon > 0$ and proper belief update mechanisms that ensure sufficient exploration of the augmented state space.

**Assumption 4 (Infinite Visitation)** *For each agent $i \in \mathcal{N}$ and each state-action pair $(s^i, u^i, b^i, a^i)$ in the augmented state space, the pair is visited infinitely often:*

$$
p\left(\sum_{k=0}^{\infty} \mathbb{I}[(s_k^i, u_k^i, b_k^i, a_k^i) = (s, u, b, a)] = \infty\right) = 1
\tag{4}
$$

*where $\mathbb{I}[\cdot]$ is the indicator function.*

---

**Algorithm 2** BQL-RM, Belief-based Q-learning with RM

---

**Input**: Agent set $\mathcal{N}$, reward machine $\mathcal{A}$, belief update function
**Parameter**: learning rate $\alpha_Q \in (0, 1)$, discount factor $\gamma \in [0, 1)$, exploration rate $\epsilon \in (0, 1)$
**Output**: Q-functions $\{Q^i\}_{i \in \mathcal{N}}$

1: **for** $i \in \mathcal{N}$ **do**
2:    $Q^i(s, u, b, a) \leftarrow$ InitQ () for all $(s, u, b, a)$
3:    $u_0^i \leftarrow$ InitRMState $(u_I)$, $b_0^i \leftarrow$ InitBelS ()
4: **end for**
5: **for** episode $= 1, \ldots, \mathcal{Q}$ **do**
6:    Initialize environment state $s_0^i$ for all $i \in \mathcal{N}$
7:    Reset RM states $u_0^i \leftarrow u_I$ for all $i \in \mathcal{N}$
8:    Reset belief states using belief update function
9:    **for** $k = 0, \ldots T$ **do**
10:      $\boldsymbol{a}_k \leftarrow \{\}; \boldsymbol{b}_k \leftarrow \{\}$
11:      **for** $i \in \mathcal{N}$ **do**
12:        *// Action selection with $\epsilon$-greedy policy*
13:        $a_k^i \leftarrow$ GetGreedyAction$(A^i)$
14:        $\boldsymbol{a}_k \leftarrow \boldsymbol{a}_k \cup \{a_k^i\}, \boldsymbol{b}_k \leftarrow \boldsymbol{b}_k \cup \{b_k^i\}$
15:      **end for**
16:      *// Environment transition*
17:      **for** $i \in \mathcal{N}$ **do**
18:        $s_{k+1}^i \leftarrow$ ExecuteAction$(a_k^i, s_k^i)$
19:        $l_{k+1}^i = L^i(s_k^i, a_k^i, s_{k+1}^i)$
20:        *// Reward machine transition*
21:        $u_{k+1}^i \leftarrow \delta^i(u_k^i, l_{k+1}^i)$, $r_k^i \leftarrow \sigma^i(u_k^i, l_{k+1}^i)$
22:      **end for**
23:      **if** mod $(k, \mathtt{m}) = 0$ **then**
24:        *// Update belief state based on observations*
25:        $\boldsymbol{b}_{k+1} \leftarrow$ BeliefUpdate$(\boldsymbol{b}_k, \boldsymbol{a}_k)$
26:      **end if**
27:      *// Update Q-values*
28:      **for** $i \in \mathcal{N}$ **do**
29:        $Q^i(s_k^i, u_k^i, b_k^i, a_k^i) \leftarrow$
30:          $(1 - \alpha_Q)Q^i(s_k^i, u_k^i, b_k^i, a_k^i) +$
31:          $\alpha_Q \left[ r_k^i + \gamma \max_{a^{i,\prime}} Q^i(s_{k+1}^i, u_{k+1}^i, b_{k+1}^i, a^{i,\prime}) \right]$
32:        $s_k^i \leftarrow s_{k+1}^i, u_k^i \leftarrow u_{k+1}^i, b_k^i \leftarrow b_{k+1}^i$
33:      **end for**
34:    **end for**
35: **end for**
36: **return** $\{Q^i\}_{i \in \mathcal{N}}$

---

Third, we assume the learning rate sequence $\{\alpha_Q(k)\}$ satisfies the Robbins-Monro conditions, ensuring appropriate convergence. Additionally, we require that the belief update mechanism provide asymptotically correct estimates of Byzantine agents.

Under the stated assumptions, we establish the following convergence result:

**Theorem 2 (Convergence of BQL-RM)** *The Q-function learned by BQL-RM converges almost surely (i.e., with probability 1) to the optimal Q-function:*

$$\lim_{k \to \infty} Q^i(s^i, u^i, b^i, a^i) = Q^{i,*}(s^i, u^i, b^i, a^i) \tag{5}$$

*where $Q^{i,*}$ is the optimal Q-function.*

The proof of Theorem 2 (detailed proof in Appendix Section BQL-RM Convergence) follows from the contraction property of the Bellman operator in the augmented state space and the Markov property preservation when beliefs are included as state representation.

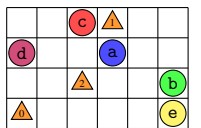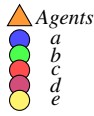

Figure 1: Grid environment showing three agents (orange triangles numbered 0, 1, 2) at their initial positions and five objects (colored circles labeled a-e) across the grid.

### 4.2 LEARNING GUARANTEES FOR BAC-RM

For BAC-RM, we use two-timescale stochastic approximation theory where critic updates occur faster than actor updates. We require Lipschitz continuity of both the policy and Q-function with respect to their parameters, ensuring stability of the gradient-based updates. Assuming appropriate two-timescale learning rates with critic updates faster than actor updates, we establish:

**Theorem 3 (Convergence of BAC-RM)** *The parameters learned by BAC-RM converge almost surely (i.e., with probability 1) to a stationary point $(\phi^{i,*}, \psi^{i,*})$ of the objective function:*

$$\lim_{k \to \infty} \|\nabla_{\phi^i} J(\phi_k^i)\| = 0 \tag{6}$$

*where $J(\phi^i)$ is the objective function, $\phi^i$ is the actor parameter, and $\psi^i$ is the critic parameter.*

The proof of Theorem 3 uses two-timescale analysis (detailed proof in Appendix Section Actor-Critic Convergence), where belief-based Byzantine detection ensures unbiased gradient estimates despite adversarial agents.

## 5 EXPERIMENTS

We evaluate our proposed algorithms, BQL-RM and BAC-RM, in a cooperative multi-agent setting with Byzantine agents using a grid-world where three agents cooperate while one provides fabricated information. The environment tests agents' ability to identify Byzantine behavior while maximizing rewards. Our second experiment (Search and Rescue) is presented in the Appendix.

### 5.1 FORAGING

We consider a foraging scenario where agents navigate a $6 \times 4$ grid-world to collect resources while dealing with Byzantine agents. This is a variation of level-based foraging (LBF) Papoudakis et al. (2020) where agents must cooperate to collect assigned resources.

Figure 3 shows results for the foraging task. Both BQL-RM and BAC-RM outperform baselines without reward machines. PPO-QMIX achieves the highest baseline performance, while COMA and M3DDPG show limited effectiveness due to the inability to capture temporal dependencies.

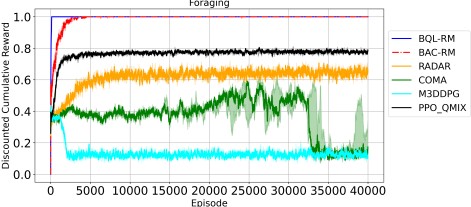

Figure 3: Cumulative rewards for the foraging task with Byzantine agents. BQL-RM outperforms baselines using reward machines and belief states.

The BQL-RM algorithm achieves higher cumulative rewards and converges faster than baselines, demonstrating the effectiveness of reward machines and belief states in Byzantine-robust c-MARL.

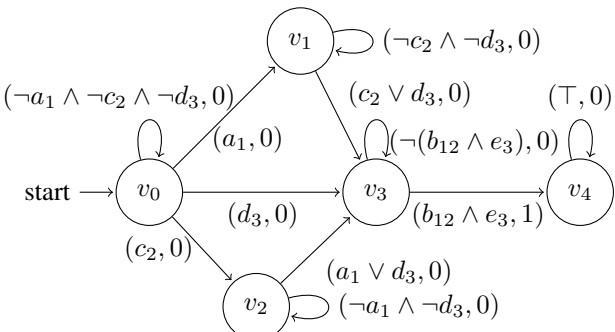

Figure 2: Team-level reward machine for the foraging task. Label $a_1$ indicates Agent 1 has visited location $'a'$, $c_2$ indicates Agent 2 has visited location $'c'$, $d_3$ indicates Agent 3 has visited location $'d'$, $b_{12}$ indicates both Agents 1 and 2 have visited location $'b'$, and $e_3$ indicates Agent 3 has visited location $'e'$. The team receives a reward only when at least two agents have completed their assigned tasks.

BAC-RM also performs well but converges more slowly, indicating both algorithms effectively learn robust policies against Byzantine agents.

## 5.2 LIMITATIONS AND DISCUSSIONS

Our experiments demonstrate that integrating reward machines with a belief-based detection mechanism provides an effective solution for temporally extended tasks in decentralized settings with local communication. Our proposed framework preserves formal guarantees in the tabular cases and provides two algorithms, BQL-RM and BAC-RM. These methods strike different trade-offs between sample efficiency and scalability, yielding faster convergence and higher cumulative returns across the evaluated benchmarks. While our algorithms perform well in the tabular setting, they may not scale well to large state and action spaces. Extending the tabular BQL-RM to settings with function approximation would enable scaling to larger state and action spaces. The current belief update mechanism compactly models agent types from observed actions, but richer inference strategies could yield more robust detection under noisy or ambiguous observations. As agent numbers increase, network constraints are likely to play a more central role, underscoring the need for communication-efficient designs. Finally, while our theoretical guarantees hold under the stated assumptions, we find that empirical performance can be sensitive to hyperparameter choices and environment dynamics.

## 6 CONCLUSIONS

We study decentralized Byzantine-resilient cooperative MARL for temporally extended tasks by using reward machines with a belief-based Byzantine detection mechanism under local agent-to-agent communication. We demonstrated that our belief update mechanism can effectively identify Byzantine agents over time, enabling robust cooperation among defenders. We introduce two classes of algorithms, Belief-based Q-learning with Reward Machines (BQL-RM) and Belief-based Actor-Critic with Reward Machines (BAC-RM) that augment the agent state with RM and belief information to isolate malicious influences. We show the convergence of BQL-RM in the tabular setting and established that BAC-RM converges to stationary points under the stated assumptions. Empirical results on a foraging grid-world and a search and rescue task shows that our methods learn robust cooperative behaviors and outperform baseline approaches in the presence of diverse adversaries. Future work will focus on scaling to large state/action spaces with function approximation, learning reward-machine structure from data Neider et al. (2021), and transferring the decentralized resilient framework to real multi-robot and networked systems Li et al. (2022).

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
