# OpenReview forum: "Decentralized Byzantine-Resilient Multi-Agent Reinforcement Learning with Reward Machines in Temporally Extended Tasks"
_ICLR.cc/2026/Conference — ICLR 2026 Conference Withdrawn Submission_

### Official Review · Reviewer_J18W · 2025-10-16

**Soundness:** 2
**Presentation:** 2
**Contribution:** 2
**Rating:** 2
**Confidence:** 4

**Summary:**

This paper provides decentralized Byzantine-robust MARL with reward machines. The authors propose BQL-RM and BAC-RM with convergence guarantees. Experiments on two tasks shows the robustness of proposed methods.

**Strengths:**

The main strength of this paper is to detect malicious agents based on the local agent-to-agent communication setting.

**Weaknesses:**

1. The author do not make a clear distinction between training-time attack and testing-time attack. In training-time attack, the task is to learn a suboptimal policy. In testing-time attack, the task is to perturb a learned, fixed policy. The author claims that "By executing the Byzantine agent’s action, the defender agents can be misled to learn a suboptimal policy" which seems like training-time attack. However, the references below includes ROMAX, M3DDPG, SA-MDP which are test-time attacks.

2. Where is the fundamental difference between reward machine and reward function and what benefit is brought by reward machine? While both are discussed, the difference is not immediately clear.

3. What is the difference between existing theoretical proof and standard Q-learning and actor-critic convergence?

4. Clarity is limited due to the excessive reference to algorithm.

5. Experiments are suspicious. First, there is only two task examined, while only one is shown in main paper. Second, what is "PPO_QMIX"? Third, since M3DDPG is a testing-time defense, I hypothesis the paper is working on this area instead of training-time attack, then [1] seems related as well. Comparison with additional, newer baselines are also welcomed.

Also, what's the difference between the proposed method and [1]?

[1] Byzantine Robust Cooperative Multi-Agent Reinforcement Learning as a Bayesian Game

1. Line 121, "improvised". do you mean "compromised".

**Questions:**

See weakness.

---

### Official Review · Reviewer_VZFr · 2025-10-30

**Soundness:** 2
**Presentation:** 2
**Contribution:** 2
**Rating:** 2
**Confidence:** 4

**Summary:**

This paper studies the problem of multi-agent reinforcement learning under adversarial attacks. The authors consider a labeled MDP model with reward machines. The authors first propose an algorithm for each agent to detect the adversarial agents among their neighbors. This is achieved by maintaining a belief state (whether a neighbor agent is adversarial or not) for each agent. Based on the adversary detection algorithm, the authors then propose a Q-learning algorithm and provide an asymptotic convergence result of the algorithm.

**Strengths:**

1. Studying resilient RL algorithms under attacks is generally important, which ensures the system performs well even under server (e.g., adversarial) environments.
2. The algorithm based on adversarial agent detection is intuitive.
3. Theoretical performance guarantees are provided for the proposed algorithms.

**Weaknesses:**

1. The adversarial model considered in the paper is not properly described or explained. While the authors claim that they consider a Byzantine adversary model, it should be noted that in such a model, each adversarial or malicious agent can send arbitrary information to its neighbors in a multi-agent system (e.g., https://proceedings.mlr.press/v206/chen23b/chen23b.pdf). However, this does not seem to be the case that the authors actually consider in the paper. For example, in the paragraph starting from line 135, the authors describe a specific way for the adversarial agent to alter/modify the original information and send it to the neighbor (rather than sending arbitrary information to the neighbors). Whether the adversarial model considered in the paper is a Byzantine model needs further clarification.
2. The consideration of the problem setup with labeled MDP model and reward machine is not well motivated or justified. Meanwhile, what are the unique challenges for designing resilient algorithms in such a problem setup, compared to standard MDP model?
3. Since the authors consider Q-learning based algorithms, they cannot be scaled to large action/state space.

**Questions:**

1. In Theorem 2 (and also Theorem 3), does the convergence result hold for all the agents $i\in\mathcal{N}$, including the adversarial agents? In addition, is Assumption 4 required for all $i\in\mathcal{N}$?
2. Could the algorithms proposed in this paper be extended to RL with function approximation, which can handle large action/state space? There is indeed a line of work on resilient RL with function approximation.
3. Could the authors further justify Assumption 4? In the standard centralized Q learning algorithm, the assumption should hold under the $\epsilon$-greedy. Does it still hold under the problem setup (decentralized with adversarial agents) considered in this paper?

---

### Official Review · Reviewer_vCDY · 2025-10-31

**Soundness:** 3
**Presentation:** 3
**Contribution:** 3
**Rating:** 4
**Confidence:** 4

**Summary:**

This paper addresses the problem of Byzantine-resilient cooperative Multi-Agent Reinforcement Learning (c-MARL) in decentralized settings with temporally extended tasks. The authors propose a framework that integrates Reward Machines (RMs) to model task structure with a belief-based mechanism for detecting malicious (Byzantine) agents. Two algorithms are introduced: BQL-RM (a tabular Q-learning method) and BAC-RM (an actor-critic method). The paper provides theoretical convergence guarantees for both algorithms and presents empirical results on a grid-world foraging task to demonstrate improved performance over several baselines.

**Strengths:**

- Novel Integration: The combination of Reward Machines for temporal abstraction with a decentralized, belief-based Byzantine detection mechanism is a novel and interesting approach. Tackling non-Markovian tasks in adversarial multi-agent settings is a relevant and challenging problem.

- Theoretical Contributions: The paper provides formal convergence guarantees for both proposed algorithms (Theorems 2 and 3). This theoretical grounding is a positive aspect, as it provides assurance about the algorithms' behavior under the stated assumptions.

**Weaknesses:**

- Fundamental Flaw in the Belief Mechanism: The proposed belief update rule (Equation 2) relies on a critical and impractical assumption: that an agent $i$ can compute the optimal action for another agent $j$. In a decentralized, partially observable setting where agents do not know each other's policies or full state, determining $a_k^{*,j}$ is fundamentally infeasible. If agent $i$ computes this using its own Q-function $Q^i$, it implicitly assumes all agents have identical optimal policies, a very strong and often invalid assumption, especially in heterogeneous tasks or during early learning. This flaw undermines the core detection mechanism and its practical applicability.

- Weak and Limited Empirical Evaluation: The experimental section is a major weakness. (1) Scale and Complexity: The evaluation is conducted only on a small grid-world with 3 agents, which does not demonstrate scalability. (2) Baseline Comparison: The baselines are not inherently designed for Byzantine resilience or temporal abstraction via RMs. Outperforming them is an expected result, not a strong validation. (3) Lack of Analysis: There is no analysis of the belief mechanism itself (e.g., precision/recall of detection). The provided figures only show cumulative reward.

- Unconvincing Practicality and Assumptions: The practicality of the approach is questionable. Assumption 1 is very restrictive. Furthermore, the assumption that defenders have a fixed policy against the Byzantine policy (Assumption 3) seems to contradict the learning process. The threat model is not tested against sophisticated adversaries that could bypass a simple action-optimality check.

**Questions:**

Please refer to weakness.

---

### Official Review · Reviewer_13sg · 2025-11-01

**Soundness:** 2
**Presentation:** 2
**Contribution:** 2
**Rating:** 2
**Confidence:** 4

**Summary:**

The paper  presents two decentralized algorithms, BQL-RM and BAC-RM, that integrate reward machines to handle temporally extended tasks with a belief based mechanism for detecting Byzantine agents. Each agent independently updates its belief about which peers may be malicious based on observed actions and rewards, allowing robust cooperation without a central controller. The authors provide theoretical convergence guarantees and demonstrate through grid world foraging and search and rescue experiments that the proposed methods outperform existing baselines by effectively identifying Byzantine agents and maintaining strong learning performance.

**Strengths:**

The paper addresses the increasingly important problem of security and reliability MARL, an area gaining attention as MARL systems are deployed in safety-critical settings such as autonomous vehicles and sensor networks. By combining Byzantine resilience with reward-machine-guided learning, the work not only strengthens the robustness of decentralized MARL but also contributes to the broader discussion of how to make cooperative AI systems secure, interpretable, and trustworthy under adversarial conditions.

**Weaknesses:**

1. Extremely limited experimental evaluation. The authors evaluate only in two case studies (small scale grid-world foraging and search-and-rescue) as per their submission.
 This makes it unclear how well the approach scales to larger, more realistic MARL domains (e.g., many agents, high-dimensional state spaces, continuous actions).

2. Simplified agent/Byzantine models.  The integrity of the belief-based detection depends on assumptions about how Byzantine agents act and what they can observe/communicate. If real adversaries operate more covertly or with richer observations, the performance may degrade. The paper does not explore a wide variety of adversarial behavior patterns.

3. Ablation studies is missing.

**Questions:**

Please compare the proposed methods against the following works:

1. Li, Simin, et al. "Byzantine robust cooperative multi-agent reinforcement learning as a bayesian game." arXiv preprint arXiv:2305.12872 (2023).

2. Bukharin, Alexander, et al. "Robust multi-agent reinforcement learning via adversarial regularization: Theoretical foundation and stable algorithms." Advances in neural information processing systems 36 (2023): 68121-68133.

Also, please include additional ablation studies, as the current experimental section is quite limited in scope.

---

### Note · Authors · 2025-11-30

I have read and agree with the venue's withdrawal policy on behalf of myself and my co-authors.